# Regulation of the Extracellular Matrix by Ciliary Machinery

**DOI:** 10.3390/cells9020278

**Published:** 2020-01-23

**Authors:** Collins I, Wann A.K.T

**Affiliations:** Centre for OA Pathogenesis Versus Arthritis, Kennedy Institute of Rheumatology, NDORMS, University of Oxford, Roosevelt Drive, Headington, Oxford OX3 7FY, UK

**Keywords:** primary cilium, extracellular matrix, transcription, secretion, endocytosis

## Abstract

The primary cilium is an organelle involved in cellular signalling. Mutations affecting proteins involved in cilia assembly or function result in diseases known as ciliopathies, which cause a wide variety of phenotypes across multiple tissues. These mutations disrupt various cellular processes, including regulation of the extracellular matrix. The matrix is important for maintaining tissue homeostasis through influencing cell behaviour and providing structural support; therefore, the matrix changes observed in ciliopathies have been implicated in the pathogenesis of these diseases. Whilst many studies have associated the cilium with processes that regulate the matrix, exactly how these matrix changes arise is not well characterised. This review aims to bring together the direct and indirect evidence for ciliary regulation of matrix, in order to summarise the possible mechanisms by which the ciliary machinery could regulate the composition, secretion, remodelling and organisation of the matrix.

## 1. Introduction

The primary cilium is a non-motile, singular organelle that is present on the surface of most mammalian cell types [1]. Cilia are assembled upon exit from the cell cycle when the “mother” centriole of the cell associates with ciliary vesicles and matures into a structure called the basal body. Accessory structures of the basal body, such as distal and subdistal appendages, facilitate docking at the plasma membrane, recruitment of proteins involved in ciliary trafficking via intraflagellar transport (IFT), and nucleation of the central microtubule core (axoneme) of the cilium [2]. The most famous role of the primary cilium is in the regulation of cellular signalling, with the cilium thought to be involved in both sensing and transducing extracellular stimuli.

Over the last 20 years, a detailed picture of the molecular processes governing cilia structure and function has been revealed through research into ciliary signalling, particularly in the context of the ciliopathies. These are genetic diseases defined by mutations affecting proteins that localise to the cilium or centrioles, known collectively as the “ciliome” [3]. Ciliopathies are characterised by defects across a wide variety of tissue types, demonstrating the importance of functional primary cilia for tissue homeostasis. These include developmental patterning defects, blindness and cystic kidneys. Formation of renal cysts is the hallmark of autosomal dominant polycystic kidney disease (ADPKD), which is the most common ciliopathy [4] and is mainly caused by mutations in the genes *PKD1* and *PKD2*, which encode the proteins polycystin-1 (PC1) and polycystin-2 (PC2). These ciliary membrane-localised proteins have been found to have a key role in mechanosensation in the kidney, and the impaired response to sensing fluid flow by the cilium is thought to bring about cellular changes that may contribute to the disease process and cystic phenotype. These changes include altered cell growth and polarity, but also changes to the extracellular matrix that lead to fibrosis [5]. The extracellular matrix (ECM) has a unique, tissue-specific composition and important functions in tissue homeostasis. For example, the ECM provides structural support, conferred by proteins such as collagen; sequesters water and growth factors due to the presence of highly charged proteoglycans; and regulates cell behaviours such as migration, proliferation, and differentiation, through non-collagenous proteins [6].

Research so far has focused on how these ECM changes affect cell-ECM interactions that regulate cell growth, morphology and adhesion, as discussed in Seeger-Nukpezah and Golemis’ 2012 review [7]. However, since then, some studies have begun to investigate more directly the link between the cilium and the extracellular matrix, and how these ECM changes may arise in the first place. Therefore, this review will focus on the recent evidence and more historic hypotheses for ciliary regulation of matrix composition, secretion, remodelling and organisation. A particular focus will be given to the tissues of the musculoskeletal system, and mainly cartilage, as this is the tissue in which the majority of research into ciliary regulation of matrix has been conducted.

## 2. Regulation of Matrix Composition

### 2.1. Ciliary Signalling Regulates Matrix Phenotype

The cilium and its associated machinery transduce a wide variety of extracellular stimuli, with the most famous example being Hedgehog (Hh) signalling. Components of the Hh signalling pathway, such as the transmembrane proteins Patched1 (PTCH1) and Smoothened (SMO), are trafficked through the cilium by intraflagellar transport (Figure 1), which is essential for regulation of the pathway [8,9,10]. Localisation of these proteins to the ciliary membrane controls activation of the pathway: in the absence of Hh ligands, PTCH1 localises to the ciliary membrane and prevents ciliary entry of SMO by an unknown mechanism. This results in proteolytic cleavage of GLI transcription factors into repressor forms that inhibit Hh target gene expression. When the pathway is activated, inhibition of PTCH1 through binding of Hh ligands enables activation of SMO and increased expression of Hh target genes through the action of full length, transcriptional activator forms of the GLI proteins [11].

Hedgehog ligands have important functions during embryonic development; for example, Sonic hedgehog (SHH) plays a key role in patterning the limb bud [12], and Indian Hedgehog (IHH), alongside other signalling molecules such as parathyroid hormone-related protein (PTHrP), regulates the differentiation of the principal cell type of cartilage, the chondrocyte, during endochondral ossification [13]. This is the process by which cartilaginous templates of skeletal elements are turned into bone, through differentiation of cartilage matrix-depositing proliferative chondrocytes into hypertrophic chondrocytes. These cells then direct the mineralisation of the entire cartilage matrix, except at articular surfaces and the growth plate. The growth plate consists of columns of proliferative, prehypertrophic and hypertrophic chondrocytes, and enables the post-natal, longitudinal growth of bone. Chondrocyte differentiation is associated with changes in the expression of matrix proteins, with proliferative chondrocytes expressing type II collagen and hypertrophic chondrocytes expressing type X collagen [14]. Modulation of Hh signalling can therefore affect the changes in matrix composition that accompany chondrocyte differentiation in the growth plate; for example, ectopic expression of *Ihh* in proliferating chondrocytes is associated with aberrant expression of type X collagen, indicative of premature hypertrophic differentiation [15]. Additionally, the matrix phenotype of the growth plate is disrupted upon direct targeting of the cilium, such as mutation of proteins involved in intraflagellar transport, which is the trafficking process responsible for assembling and maintaining the cilium [16]. Cartilage-specific knockout of KIF3A, which is a motor protein that transports IFT particles containing cargo from the cell body, is associated with altered type X collagen expression in the mouse growth plate [17]. Reduced proteoglycan production and type X collagen expression are also observed upon deletion or mutation of proteins that form the IFT particles, such as IFT80 [18] and IFT88 [19]. Therefore, the growth plate can be used as an exemplar of how ciliary signalling can determine matrix composition as a function of Hh-regulated differentiation.

However, the exact mechanism by which the Hh pathway influences the expression of matrix genes is unclear, as is the role of Hh signalling beyond development in post-natal cartilage and, more generally, in adult tissues. One possible mechanism is the direct regulation of matrix gene expression by GLI transcription factors. Through supervised analysis of Hh-responsive GLI binding targets in neoplastic chondrocytes, Ali et al. identified multiple ECM genes [20], indicating the extent of potential matrix regulation either directly or indirectly downstream of ciliary Hh signalling. Given the identification of TGFβ signalling targets such as *BMP2* as GLI target genes, these results also indicate that there is complex co-regulation of matrix genes by Hh and other signalling pathways. Thus, changes to matrix composition observed upon disruption of cilia genes could also be attributable to signalling beyond Hh, particularly as KIF3A knockout mice exhibited no change in *Ptch1* mRNA expression [17]. The authors of Haycraft et al. [21] also point out that not all changes induced by IFT88 deletion match those seen in Hh mutant mice, such as *Ihh;Gli3* double mutants. Additionally, as highlighted in both Song et al. [17] and McGlashan et al. [19], disruption of these ciliary genes results in changes to cell organisation and morphology, most notably loss of chondrocyte columnar orientation. In KIF3A knockout mice, this altered orientation was also associated with changes in localisation of focal adhesion kinase (FAK). Furthermore, both KIF3A knockouts and IFT88 mutant mice exhibit changes to the actin cytoskeleton, thus providing further evidence for a link between ciliary proteins and the cytoskeleton that appears to be conserved across many cell types. These studies propose that such tissue and cell patterning changes, and associated matrix changes, are due to disruption of the Wnt pathway, including non-canonical planar cell polarity (PCP) (see Section 5). Yuan et al. suggested that the changes downstream to loss of IFT80 were due to altered chondrocyte differentiation, as a result of reduced Hh signalling and increased Wnt signalling [18]. Similarly, Serra et al. attributed growth plate changes observed upon constitutive deletion of IFT88 to changes in Wnt signalling downstream of dysregulated ciliary Hh signalling [22]. Therefore, the regulation of cell and matrix phenotype by the ciliary machinery is likely controlled through a number of mechanisms including Hh signalling, Wnt signalling, and cytoskeletal rearrangement. It is possible, and has previously been hypothesised, that interpretation of tissue mechanics by chondrocytes ultimately regulates all these behaviours, as discussed in the next section, but this has yet to be proven experimentally.

### 2.2. Mechanical Forces Regulate Matrix Composition through Ciliary Signalling

Intracellular signalling pathways that modulate expression of matrix molecules, including the Hh pathway, may also be activated by changes in the mechanical load experienced by a cell [23]. The cilium is proposed to have a crucial role in sensing such forces and transducing mechanical stimuli into a cellular response. This role for the cilium in reception and transduction of mechanical stimuli was proposed as early as the 1970s and 80s, for example by Poole et al., who observed bending of the cilium in isolated chondrocytes and tendon cells [24]. Praetorius and Spring later observed an increase in intracellular calcium concentration upon bending of cilia on the apical surface of kidney epithelial cells, indicating that primary cilia are mechanically sensitive [25]. This calcium response is thought to be mediated by polycystin proteins (PC1 and PC2) which form a complex that localises to the ciliary membrane [26] (Figure 1).

Ciliary localisation of the PC1/PC2 complex is required for mechanosensitive calcium signalling: loss of cilia in mice harbouring a hypomorphic mutation affecting the IFT protein IFT88 resulted in a reduced calcium response to fluid flow [27]. These mice, known as ORPK mice, are used as a model of autosomal dominant polycystic kidney disease (ADPKD) [28]. This is one of the most common ciliopathies and is most frequently caused by mutations affecting PC1 in humans [29]. ADPKD is defined by a cystic kidney phenotype, which is associated with a number of cellular changes: kidney epithelial cell growth and proliferation; loss of cell polarity; and also altered expression of extracellular matrix proteins, such as increased expression of collagens, that leads to fibrosis [5]. Mangos et al. showed that knockdown of PC1 during zebrafish development resulted in collagen overexpression in various tissues, linking PC1 loss of function to expression of ECM components [30]. However, PC1-deficient zebrafish exhibited reduced collagen expression in the embryonic kidney, but also a mild cystic kidney phenotype. Therefore, whilst fibrosis has been shown to correlate with cyst severity [31], and other factors such as increased cell proliferation are not the primary cause of cyst formation in some disease models [32], the contribution of such fibrotic changes to cyst initiation during ADPKD pathogenesis is not yet fully understood, and is reviewed in more detail elsewhere [7,33].

Studies in cartilage have given further insight into the mechanisms by which the cilium regulates matrix molecule expression in response to mechanical forces. Wann et al. observed an increase in the expression of the key cartilage matrix component aggrecan, downstream of compression-induced calcium signalling [34]. This upregulation in aggrecan expression was not observed in chondrocytes derived from the ORPK mouse, which also exhibited a loss of calcium signalling. However, disruption of the cilium in ORPK chondrocytes did not alter the purinergic signalling events, previously reported in the kidney [35], that occur upstream of calcium signalling and have been shown to regulate proteoglycan synthesis in chondrocytes [36] (Figure 1). Instead, the group proposed that altered expression of PC1, which has been shown to influence ATP-mediated calcium signalling [37], resulted in ORPK chondrocytes failing to transduce a purinergic signal to the appropriate calcium signal required for a normal anabolic matrix response to mechanics. Specifically, ORPK chondrocytes did not express full-length PC1, but only smaller ~35kDa protein that was possibly a C-terminal cleavage product of the full-length protein. In addition to IFT88, KIF3A has been shown to potentially regulate mechanosensitive matrix expression. Rais et al. found that knockdown of KIF3A in a chondrocytic cell line altered the transcriptional response to mechanical stimulation, including the regulation of genes encoding aggrecan, type II collagen, and type X collagen [38]. The authors also found that mechanical loading altered Hh signaling and increased ciliation in the chicken growth plate in vivo.

Fibrosis in non-ciliopathies has also been linked to the primary cilium. In investigating the mechanisms underlying fibrosis after cardiac injury, Toomer et al. found that conditional knockout of IFT88 in adult mouse heart leads to increased expression of ECM components, including the proteoglycan versican, and formation of immature collagen fibres, indicating involvement of a post-transcriptional defect [39]. Also, Villalobos et al. found evidence of a relationship between TGFβ-SMAD2/3 signalling, collagen expression and cilia in the context of cardiac fibrosis [40]. Horn et al. showed that another fibrotic disease, systemic sclerosis, involves activation of the Hh pathway [41]. The authors also found that treatment of healthy human fibroblasts with SHH resulted in increased type I collagen expression and ECM release from the cells in vitro, and overexpression of SHH in mice led to the induction of dermal fibrosis. Genetic studies have also linked fibrosis to the cilium in diseases such as idiopathic pulmonary fibrosis [42] and congenital heart disease [43], but mechanistic evidence is lacking.

Another tissue which exemplifies how modulation of cilia function by mechanical forces can alter cell behavior is the vasculature, where high levels of fluid flow are proposed to inhibit ciliary assembly and/or result in cilia disassembly [44]. This affects cellular processes such as mechanosensation, with non-ciliated endothelial cells unable to initiate calcium signalling in response to fluid shear stress [45]. Even when cilia are assembled, their function is altered, with potentially pathophysiological consequences in the context of diseases such as atherosclerosis [46,47]. How cilia may interact with the matrix at the vascular endothelium, such as the glycocalyx, is, to our knowledge, not understood.

## 3. Regulation of Matrix Secretion

In order to achieve the complexity necessary for the highly specific functions of the ECM, cells exert control over matrix composition beyond transcription of genes encoding matrix molecules. The cell must also secrete large and often highly modified ECM molecules, and there is some evidence to suggest that the cilium is also involved in this process.

### 3.1. Primary Cilia Are Proposed to Act in a Matrix-Cilium-Golgi Continuum

The idea that the primary cilium is involved in secretion of matrix components was first explored by Poole et al. This group observed an ultrastructural relationship between the cilium and the Golgi in cells within different load-bearing and non-load-bearing tissues, such as chondrocytes, tenocytes and aortic smooth muscle cells, from a number of different species [24,48]. Electron microscopy (EM) analysis of these tissues found that the *trans* face of the Golgi, together with numerous vesicular structures, were closely associated with the basal body and base of the cilium. The proximity of the Golgi to the cilium led the authors to hypothesise that there is a relationship between the two structures and the matrix. Further support for this hypothesis came from the same group, who observed interactions between the ciliary axoneme and matrix components such as aggregated proteoglycans and type II collagen fibres [49], possibly mediated by integrins later found to be present on the ciliary membrane [50]. This series of interactions was termed the matrix-cilium-Golgi continuum. In the continuum, cilia sense mechanical changes in the matrix and transduce this information to the cell body. This results in the required changes in matrix secretion, which are polarised to the region around the cilium, as facilitated by the close proximity of the cell’s secretory machinery. There is no direct evidence for this, but a few subsequent studies have revealed how different parts of the continuum may be linked.

### 3.2. IFT20 Links the Golgi to the Cilium

One aspect of the continuum for which there is substantial evidence is the linking of the Golgi with ciliary IFT machinery, principally through the actions of IFT20. This protein has been shown to traffic along the ciliary axoneme as part of the IFT particle, and thus IFT20 knockout inhibits cilia assembly in retinal epithelia [51] and conditional IFT20 deletion in mouse kidney disrupts ciliogenesis and leads to cystic kidney disease [52]. IFT20 has also been shown to colocalise with markers of the *cis*-Golgi, where it is anchored by the protein GMAP210 [51,53] (Figure 1). Since depletion of IFT20 resulted in reduced ciliary localisation of the transmembrane protein polycystin-2 in vitro and in vivo, these findings led to the hypothesis that IFT20 at the Golgi is involved in sorting cargo destined for the primary cilium.

Subsequent studies have provided further insight into the mechanism by which IFT20-mediated transport of ciliary cargo occurs. Given the strong colocalisation of IFT20 with markers of the *cis*- rather than *trans*-Golgi, where proteins are typically packaged into vesicles for delivery to target structures, Gonazlo-Garcia and Reiter suggested that IFT20 tags ciliary proteins early in the Golgi secretory pathway, then “hands over” cargo to components of the Arf4 transport machinery [54]. These include Arf4 itself, which mediates formation of cargo-containing vesicles, and Rab11 and Rab8, which are recruited by Arf4 to direct transport to and fusion with the periciliary membrane respectively [55]. Further support for the existence of two separate pools of IFT20 (one at the ciliary base, for IFT, and one at the *cis*-Golgi), rather than continuous transport of IFT20-containing vesicles between the two compartments, comes from the observation that Golgi trafficking of IFT20 is not required for IFT20 localisation at the centrosome [53]. However, without live cell microscopy of both IFT20 and ciliary cargo-containing vesicles, it is not possible to rule out the latter hypothesis.

In either case, IFT20 is likely involved in transport of ciliary proteins from the Golgi, which are then delivered to the ciliary membrane by exocytosis once their host vesicles arrive at the ciliary base (Figure 1). Evidence for exocytosis occurring in this region comes mainly from study of the exocyst, an eight-protein complex that, together with the Rab11/8 pathway, facilitates docking and fusion of post-Golgi secretory vesicles with the plasma membrane [56]. One of the components of the exocyst, Sec10, has been shown to localise to cilia in vitro in MCDK cells via EM [57] and in vivo [58]. Sec10 also interacts with PC2, and morpholino-mediated Sec10 knockdown in zebrafish causes ADPKD phenotypes together with reduced ciliary membrane PC2, although this could be secondary to the severe ciliogenesis defect also observed [59]. Examples of proteins that are incorporated into the ciliary membrane after transport through the Golgi-localised IFT20 pathway include opsin, which is a G protein-coupled receptor that is crucial for phototransduction in photoreceptor cells [60].

Therefore, the mechanism by which IFT20 mediates transport of transmembrane ciliary proteins to the ciliary region, leading to their subsequent exocytosis, is relatively well characterised. IFT20 has also been shown to have a role in the intracellular trafficking and secretion of other proteins, including extracellular matrix proteins. Noda et al. found that conditional knockout of IFT20 in mouse cranial neural crest cells, where it is required for cilia assembly, also led to delayed secretion of type I collagen during development [61]. Reduced colocalisation of type I collagen with the *cis*-Golgi marker GM130 and impaired trafficking of VSVG-GFP (a widely used protein for studying intracellular transport) between the ER and Golgi, indicated that IFT20 plays a role in ER-to-Golgi transport of type I collagen. GMAP210 is known to have a role in this process, but Smits et al. found that mutation of GMAP210 in mice does not result in impaired ER-to-Golgi trafficking, nor collagen defects [62]. These observations possibly rule out GMAP210-related defects in the collagen secretion phenotype, despite this mutation also resulting in impaired glycosylation and secretion of non-collagenous matrix molecules, namely perlecan. Kitami et al. found that IFT20 knockout in adult mouse cartilage leads to loss of cilia, Hh signalling defects, and, in contrast to developmental IFT20 deletion, Golgi morphological changes due to altered input and output of cargo [63]. These morphological changes and decreased secretion are also observed in GMAP210 mutant patient-derived cells, indicating that these defects may be related more to changes in global secretory load, than specific, IFT20-mediated, trafficking from the Golgi [64].

If these defects arise due to disturbance of IFT20 function in sorting ciliary-destined proteins early in the Golgi, and thus facilitating their transport to the periciliary region, why might matrix molecules also be sorted in this way, and use a pathway that ends in exocytosis at the cilium for their secretion? To understand this, live cell imaging of collagen and other matrix molecule transport would be required. Therefore, there is limited evidence for IFT20 having a role in mediating secretion of matrix molecules, and even less evidence for *polarised* secretion of matrix components at the cilium. However, ciliary proteins are known to mediate polarised secretion in another context, which is addressed in the following section.

### 3.3. Polarisation of Secretion Occurs at the Immune Synapse

IFT proteins including IFT20 also have roles in intracellular transport in immune cells such as T lymphocytes, which do not assemble a cilium but undergo extensive trafficking activity upon formation of the immune synapse. This structure is assembled when a T cell encounters its target cell and activates T cell receptor signalling. This in turn elicits a series of cellular events, including polarisation of the centrosome at the plasma membrane in a similar way to the docking of the basal body during ciliogenesis [65,66]. This is followed by polarised secretion of cytotoxic granules at the immune synapse, which is facilitated by specific changes in the phosphoinositide composition of the plasma membrane, as demonstrated by Gawden-Bone et al. [67]. The final composition of the immune synapse is similar to that of the ciliary membrane, with both bilayers exhibiting low levels of phosphatidylinositol 4,5-bisphosphate (PI(4,5)P_2_) and high levels of phosphatidylinositol 4-phosphate (PI(4)P) [68,69]. Therefore, the specific phospholipid composition of the immune synapse and potentially the primary cilium may provide a specialised membrane domain for secretion; however, there is currently no evidence to suggest that modulation of this composition alters secretion in the context of cilia.

Further evidence for polarisation of exocytic and vesicular trafficking processes towards the immune synapse comes from studies of T cell receptor recycling. Das et al. found that both the T cell receptor and transferrin receptor, which is constitutively endocytosed and recycled back to the plasma membrane, are polarised at the synapse upon establishment of contact between a T cell and an antigen presenting cell (APC) [70]. Subsequent studies found that IFT20 is required for trafficking of T cell receptor (TCR) clusters and the co-receptor CD3 to the immune synapse [71], and specifically in trafficking between the early endosome (the site of sorting of internalised cargo) and the recycling endosome [72]. Onnis et al. showed that disruption of Rab29 results in decreased ciliation together with a reduction in the number of vesicles present at the ciliary pocket, which is a specialised membrane domain at the base of the cilium that is a site of endocytic activity [73]. The same paper also showed disturbed trafficking of TCR-containing endosomes along microtubules towards the immune synapse in a process similar to IFT. This work provides further support for the hypothesis that both the cilium and the immune synapse represent “specialised membrane patches” [71] that are formed through vesicular transport from the Golgi to the plasma membrane to define a new membrane domain.

Therefore, there is evidence that IFT20 mediates polarised secretion, including towards structures that are structurally and biochemically similar to the cilium and periciliary region. However, there are still many open questions to investigate in the context of the continuum, and in the polarised secretion of matrix molecules specifically. In light of the evidence that the phospholipid composition of both the ciliary membrane and the immune synapse is distinct from the rest of the plasma membrane, how does this composition at the cilium affect matrix molecule secretion? To our knowledge, there is no direct evidence that shows a role for IFT20 itself in the modulation of membrane composition, only that Golgi-to-cilia trafficking of IFT20 is impaired upon mutation of the Golgi-localised protein phosphoinositide 3-kinase regulatory subunit 4 (PIK3R4), which regulates the kinase that produces phosphatidylinositol 3-phosphate (PI(3)P) [74]. It is more likely that any perturbation of cilia assembly, or wider trafficking occurring in the periciliary or immune synapse region, disrupts the localization of phosphoinositide-regulating enzymes and other effectors. For example, Kosling et al. suggested that ciliary localization of the phosphatase INPP5E is IFT-dependent, which could thus affect ciliary membrane composition (discussed above) [75]. However, whether the ciliary pocket membrane has a unique composition, and whether this could be affected by failure of axonemal, “innerciliary” trafficking of enzymes such as INPP5E, is not well understood. Another question yet to be addressed in the literature is, if these polarised secretory pathways exist at the cilium, is it just secretion of cartilage matrix components that is polarised, or does the cilium represent a global site of polarised secretion? What would be the evolutionary benefit of this? Can the ciliary protein-targeting machinery cope with transport of molecules as large as matrix components?

## 4. Regulation of Matrix Degradation

Another crucial process involved in the maintenance of ECM homeostasis is the degradation of matrix macromolecules in response to changes in the extracellular environment. For example, this process is essential in the migration of cells and remodelling in response to mechanical loads, enabling adaptation of the mechanical properties of the tissue. The main mechanism by which matrix components are degraded is through the actions of proteases, specifically metalloproteinases. The cilium and periciliary membrane domain have been linked to the regulation of the expression and activity of these enzymes in a number of ways, as discussed below.

### 4.1. Ciliary Signalling has been Linked to Transcriptional Control of Protease Expression

A number of studies have looked at the role of the cilium and the signalling pathways it modulates, in regulating the expression of matrix-degrading proteases. Many of these studies have focused on this role in the context of osteoarthritis (OA), a disease defined by increased protease activity that leads to excessive degradation of the cartilage extracellular matrix. There are currently no disease-modifying drugs to treat OA, and therefore identifying the mechanisms by which these enzymes are regulated, and also dysregulated within the disease process, is of great clinical interest.

In OA, adult chondrocytes exhibit hypertrophic-like changes to the healthy phenotype, such as increased production of the collagenase matrix metalloproteinase 13 (MMP-13) [76]. During post-natal development, hypertrophic differentiation of chondrocytes, which is regulated by Hh signalling, is an important step in the formation of bone from a cartilage template by endochondral ossification, as discussed in Section 2.1. In the adult context, these hypertrophic changes switch from being essential for the matrix remodelling that facilitates subsequent developmental processes, to being pathological, resulting in excessive matrix degradation that compromises the structural integrity of the tissue. These observations, together with evidence of primary cilia length and incidence changes in large animal models of OA [77], formed the basis for subsequent investigation into whether the OA hypertrophic phenotype (and thus excessive matrix degradation) was related to aberrant activation of Hh in adult cartilage. Lin et al. found that *Ptch1*^−/−^ knockout mice exhibited cartilage changes consistent with those seen in OA tissue, such as loss of proteoglycan staining and upregulation of *Mmp13* and the gene encoding a protease called A Disintegrin and Metalloproteinase with Thrombospondin motifs 5 (ADAMTS-5, which degrades a key component of cartilage, aggrecan), in addition to upregulation of Hh target genes [78]. An even more severe phenotype was observed with cartilage-specific activation of SMO induced in adult mice, whilst genetic and pharmacological inhibition of the pathway after surgical induction of OA reduced disease severity. A later study reported that conditional knockout of IHH, the Hh ligand that directly stimulates hypertrophic differentiation in the cartilage growth plate, decreased MMP activity, reduced cartilage damage, and reduced catabolic and increased anabolic biomarker production, relative to wild-type (WT) mice [79]. Lin et al. also found that Hh signalling targets were upregulated in human OA samples, and expression of *MMP13* and *ADAMTS5* were increased upon stimulation of human cartilage explants with Hh ligand. This response was abolished upon knockdown of RUNX2, the master transcription factor regulating chondrocyte hypertrophic differentiation [80], which directly interacts with the *ADAMTS5* gene promoter [81]. *IHH* levels have also been found to correlate with OA disease severity and chondrocyte size in human OA tissue [82].

In addition to these studies investigating Hh signalling specifically, there is also evidence that disruption of the cilium at the level of the organelle results in changes in matrix degradation in adult cartilage. Genetic disruption of the BBSome, which controls membrane trafficking to the cilium and is defective in the ciliopathy Bardet-Biedl syndrome, resulted in cartilage matrix changes in adult mice [83]. Knockout of the BBS proteins BBS2 or BBS6, or mutation of BBS1, resulted in complete loss of superficial articular cartilage and reduced proteoglycan content in the remaining cartilage. In contrast, analysis of the BBS1 mutant at earlier time points showed that mutant mice have thicker cartilage relative to WT, but also have areas of articular surface erosion [84]. Similarly, cartilage-specific deletion of IFT88 resulted in increased cartilage thickness and staining for type II collagen and proteoglycans, but also upregulation of *Adamts5*, *Mmp13* and *Runx2* expression, and Hh signalling [85]. This increased anabolic response may be consistent with the attempts at repair seen during the early stages of OA, but the authors propose that more work is required (specifically the use of inducible knockdown of ciliary proteins in adult mice) to address whether these defects are related to failure to synthesise healthy cartilage matrix in the first place, or related instead to increased catabolism during adulthood. Therefore, it is likely that there is a combination of both anabolic and catabolic dysregulation in these mouse models.

A number of studies have investigated the molecular processes underlying the results discussed above. For example, in addressing why Hh signalling is increased in human OA, Thompson et al. focused on the role of mechanics [23], which has previously been shown to modulate the expression of matrix-degrading proteases in various cell types such as fibroblasts [86], and of *IHH* [87]. They found that cyclic tensile strain applied to bovine chondrocytes results in activation of the Hh pathway, mediated by the primary cilium, and increased expression of *ADAMTS5*. These results led to the hypothesis that mechanics-stimulated induction of the Hh pathway results in expression of proteases such as ADAMTS-5 and MMP-13 via RUNX2 (Figure 1). This was supported by observations that GLI2 interacts with RUNX2 [88], knockdown of RUNX2 prevented upregulation of *ADAMTS5* and *MMP13* in response to strain in vitro [89], and *Mmp13* expression and cartilage damage were reduced in a mechanically induced OA mouse model [90]. However, direct stimulation of chondrocytes, in vitro or in explants, with IHH did not result in increased matrix degradation or protease expression [91]. The authors proposed that this difference may be due to the use of short time courses, which may mask any effects on long-term chondrocyte fate. These longer-term effects may explain the phenotype of mice with genetically modulated Hh signalling observed by Lin et al. [78], particularly in light of the role of Hh in chondrocyte differentiation during development. Another important factor which could help to explain these contrasting results obtained through the strain studies in vitro and mouse models in vivo, and experiments using isolated cultured cells and tissue, is mechanics. Forces experienced by the chondrocyte in loading experiments and in vivo will be transduced to the cell cytoskeleton and in turn could affect fundamental cellular processes. These include trafficking between the cytosol and the nucleus, and ciliogenesis itself, as cilia have been shown to disassemble under mechanical load [92]. Such cell behaviour-modifying forces are absent in non-loaded cells and isolated tissue, and should therefore be considered when interpreting results from experiments using these systems. Further work is also required to better understand how cell-intrinsic signalling pathways such as Hh integrate with extrinsic mechanical cues.

As the experiments in Thompson et al. [91] were conducted using healthy bovine chondrocytes or cartilage explants, the authors also hypothesise that Hh signalling may be exerting an effect on protease expression and matrix degradation through interactions with another factor that is only present in the disease state. To address this, they focused on the IL-1β pathway, as inflammatory cytokines including IL-1β are present in the OA joint [93] and IL-1β itself is a strong inducer of protease-mediated cartilage matrix catabolism [94]. However, treatment of chondrocytes with IHH or the Hh antagonist cyclopamine, alongside IL-1β stimulation, did not affect IL-1β-induced matrix degradation, despite there being evidence of cross-talk between the two pathways. 

Subsequent studies have looked at other pathways that may interact with Hh signalling to regulate protease expression. Since Hh and Wnt signalling have important roles in joint morphogenesis during development and interact in various other tissues, the interaction between Hh and Wnt pathways has also been studied in the context of OA. Rockel et al. found that expression of selected Wnt pathway target genes, including *ADAMTS5* and *MMP13*, is modulated by Hh-induced negative regulators [95]. Perhaps the most promising Hh-regulated pathway is cholesterol biosynthesis, which was identified as the most significantly dysregulated pathway at the transcriptional level in human cartilage treated with Hh ligands or antagonists [96]. Cartilage-specific knockout of the main negative regulator of cholesterol biosynthesis, INSIG1, led to increased cholesterol accumulation, cartilage damage and increased expression of *Mmp13* and *Adamts5*.

Apart from this study [96], there is still very little direct, mechanistic evidence for how the Hh pathway, either independently or through interaction with other pathways, regulates matrix degradation. To obtain better insight into the mechanism of action of Hh in matrix catabolism, it could be helpful to look beyond protease expression, and instead towards how ciliary signalling may regulate the activity of matrix-degrading proteases in other ways, such as through control over their endogenous inhibitors. For example, TIMP-3 inhibits the collagenase MMP-14/MT1-MMP, and is therefore involved in remodelling the type I collagen matrix surrounding preadipocytes and thus modulating cell-matrix interactions [97]. This function of TIMP-3 is particularly important in directing the normal repair process after muscle injury, which is mediated by muscle-resident mesenchymal cells known as fibrogenic/adipogenic progenitors (FAPs). Kopinke et al. found that FAP ciliation decreases during the injury response, reducing Hh signalling and resulting in the derepression of the Hh target *Timp3* [98] (Figure 1). As TIMP-3 has been shown to be important in regulating protease activity in vitro [99] and in vivo [100], and is present at reduced levels in OA relative to normal tissue [101], investigating the TIMP-3-Hh relationship in cartilage may provide a new direction for investigation of the mechanism of Hh-regulated protease activity.

### 4.2. Primary Cilia have been Associated with Endocytic Control of Protease Activity

There is some evidence other than regulation of TIMP-3 by Hh signalling, that the cilium has a role in the post-transcriptional regulation of proteases. Studying this mechanism of regulation may be of particular interest since there is evidence that protease expression does not always correlate with cartilage degradation. For example, expression of the gene encoding ADAMTS-5, which is the most active aggrecan-degrading protease in vitro [102], has been found to be only slightly upregulated in OA tissue relative to other proteases [103], or unchanged [104]. Therefore, altered transcription may not be the principal method of dysregulation of protease activity, which may instead be at the post-translational level. This mechanism is thought to be the rapid clearance of secreted proteases from the extracellular matrix by receptor-mediated endocytosis, enabling regulation of the activity of potent matrix-degrading enzymes over a much shorter time period than transcriptional control.

MMP-13 and ADAMTS-5 have been shown to bind to the cell surface receptor LRP-1, and the resulting receptor-protease complex is internalised via clathrin-mediated endocytosis, facilitating the targeting of the protease to the lysosome for degradation [105,106,107]. Research in our own lab showed that LRP-1-mediated endocytosis of proteases is impaired in chondrocytes derived from the ORPK mutant mice. Co-culture of these cells with the purified matrix component aggrecan, and subsequent detection of protease-generated aggrecan neoepitopes, showed that ORPK chondrocytes have constitutively increased protease activity in vitro [108]. This increased activity was not associated with increased Hh signaling or increased mRNA levels of aggrecan-degrading proteases. We also observed polarisation of internalised proteases and LRP-1 itself at a region of the cell surface around the cilium (Figure 1). This polarisation was observed less frequently in ORPK chondrocytes, leading to the hypothesis that the primary cilium is a “hotspot” for the endocytosis of proteases. Endocytic uptake of other ADAMTS proteases at the cilium has been observed in other cell types. Nandadasa et al. found that secreted ADAMTS-9, which is involved in remodelling the embryonic ECM, localises to Rab-11-positive late recycling endocytic vesicles surrounding the basal body [109]. These vesicles are generated by clathrin-dependent LRP-1/2 receptor-mediated endocytosis, and ADAMTS-9 was also found to be required for ciliogenesis through an unknown mechanism that may be linked to the formation of the ciliary vesicle, which involves Rab11, Rabin8 and other endocytic proteins involved in the late endocytic recycling pathway [55,110,111].

Many other studies have reported endocytosis occurring at the cilium, and particularly at the ciliary pocket. This is a pit-like structure formed through invagination of the plasma membrane surrounding the base of the cilium (Figure 1). The ciliary pocket is morphologically similar to the flagellar pocket found in the protozoan parasite trypanosomes, which is the exclusive site of endocytosis and exocytosis across the cell surface [112]. Molla-Herman et al. first demonstrated endocytosis occurring at the ciliary pocket, with clathrin-coated vesicles at the ciliary pocket membrane observed via electron and immunofluorescence microscopy [113]. A number of studies have since provided an insight into the function of these endocytic processes at the cilium. In addition to regulating ciliogenesis, endocytic proteins have been shown to be involved in regulation of signalling by the cilium, as illustrated by Bhattacharyya et al. who found that the knockout of the endocytic recycling protein EHD1 results in altered primary cilia morphology and increased SHH signalling [111]. Additionally, Schou et al. found that the kinesin KIF13B is required for enrichment of the protein caveolin-1 at the transition zone [114]. Caveolin-1 (CAV1) is involved in formation of caveolae, which are types of lipid microdomains in the plasma membrane that are involved in endocytosis. Given that both KIF13B and CAV1 knockdown reduced ciliary SMO localisation upon SHH treatment, these results further support a role for endocytosis in the trafficking of Hh pathway components. Regulation of cell signalling is also associated with endocytosis at the cilium in the context of TGFβ signalling: the TGFβ receptor I colocalises with clathrin, endocytic adaptor proteins, and early endosome markers at the ciliary pocket, with downstream activation of SMAD2/3 reduced both in ORPK mutant cells, and upon pharmacological inhibition of endocytosis [115].

More work is required to determine the exact mechanism by which the cilium may regulate the endocytosis of proteases, such as determining whether ciliary proteins interact with the endocytic machinery. KIF13B has already been shown to interact with LRP-1 and regulate its caveolae-mediated endocytosis [116]; however, endocytosis of proteases is mediated by clathrin-dependent internalisation. Also, further investigation is required into how the apparent polarisation of endocytosis at the cilium, such as that observed for ADAMTS-5 and ADAMTS-9, is set up. In trypanosomes, endocytosis is restricted to the flagellar pocket due to the presence of a densely packed microtubule network beneath the plasma membrane throughout the rest of the cell [112]. Such a structure does not exist in cells assembling a cilium, which is the preferential rather than the exclusive site for endocytosis in some contexts, suggesting that any possible polarisation of trafficking activity at the cilium is established in a different way to the flagellar pocket. One possible hypothesis is that the properties of the periciliary environment may enhance the endocytic capacity in this region, specifically the phosphoinositide make-up of the ciliary and ciliary pocket membranes. This is different from the rest of the cell membrane (as discussed in Section 3.3) [68,69], and there is also evidence that the lipid composition of the flagellar pocket membrane affects endocytosis [117]. However, it is still unclear what benefit the polarisation of endocytic processes provides to the cell. Speculatively, the function of such polarisation could be related to the role of the cilium in mechanotransduction. Endocytosis has previously been linked to the cellular response to mechanics: clathrin-mediated endocytosis was enhanced in response to fluid flow-induced shear stress in renal proximal tubule cells through a cilia-dependent increase in calcium signalling [118], and has also been shown to mediate the polarised uptake of proteases, the activity of which is also regulated by mechanics. Therefore, the close proximity of mechanotransduction events and potentially mechanosensitive endocytic processes at the primary cilium could facilitate more effective cellular responses to such forces. A potential link between mechanosensation and endocytosis is the actin cytoskeleton, as the ciliary pocket is a site of actin filament docking [113], and actin regulates various stages of the endocytic pathway [119]. Approaches that could be used to test this hypothesis include development of experimental techniques targeting the ciliary pocket directly; comprehensive study of interactions of ciliary proteins with endocytic proteins; and investigation into how cilia-polarised endocytosis may enhance uptake of cargo, namely through increased rate of internalisation, transport through the endosomal pathway, or recycling to the plasma membrane.

## 5. Regulation of Matrix Organisation

As discussed in the above sections, the specific composition of the extracellular matrix is determined by transcriptional programs, secretion of matrix components and their degradation by proteases, and this composition in turn defines the functions of the matrix. Another important factor that contributes to establishing the properties of the matrix is the spatial organisation of these matrix components. In many polarised tissues, such as the kidney and vasculature, the cilium extends into the lumen of tubules and vessels due to its assembly on the apical side of epithelial cells, and thus on the opposite side to the majority of the matrix, which is secreted and remodeled predominantly on the basolateral surface of cells. It is not clear how the ciliary machinery could directly participate in organizing these matrices. However, in tissues where cilia are in direct contact with the matrix, it is possible that the cilium could have such a role. The spatial organization of matrix is well-illustrated in compressive or tensile load-bearing tissue, where the matrix is highly spatially, and often anisotropically, organised. For example, the orientation of type II collagen in the cartilage matrix relative to the articular surface varies throughout the different layers of the tissue [120], first described by Benninghoff as “arcade”-like structures almost 100 years ago [121]. The highly organised nature of the type II collagen network is thought to determine the biomechanical properties of cartilage, given that there is a correlation between collagen network organisation in the superficial zone with stiffness [122], and computational modelling of deep and superficial fibrils indicated that these networks protect against tensile strain across the whole matrix and articular surface, respectively [123]. Despite its functional relevance in relation to the biomechanical properties of cartilage, how this complex spatial arrangement is established is not well characterised. It is not likely to be due to an intrinsic property of collagen molecules, which can self-assemble into fibrils but do so without any spatial organisation [124]. Mechanical forces have been shown to be necessary for establishing this proper collagen architecture, with loading in intact rabbit knee joints changing the collagen fibre alignment in physiologically unloaded regions of cartilage [125]. Also, studies in porcine [126] and ovine cartilage [127,128] have shown that collagen fibril orientation changes from birth, when almost all fibrils are organised parallel to the articular surface, to maturity, when arcade-like structures are fully established. During this time, cartilage is subjected to increasing levels of mechanical force. Therefore, these results indicate that there may be an additional cellular regulator that helps establish the complex orientation of matrix components during post-natal development, possibly through sensing and transducing mechanical forces.

One of the hypotheses for how tissue anisotropy is established in cartilage focuses on the primary cilium, as described by Farnum and Wilsman [129]. This hypothesis is that the cilium senses the mechanical forces exerted on the local microenvironment of a chondrocyte. The ciliary machinery subsequently communicates to the cell where the downstream cellular processes that facilitate adaptation to these forces should be directed, thus generating region-specific differences in matrix organisation. These processes include polarised secretion of matrix molecules, following the observations of a spatial, and later molecular, interaction between the Golgi and the cilium (matrix-cilium-Golgi continuum, as discussed in Section 3.1); regulation of cell shape, through association with the microtubule network in the cell; and in light of the more recent evidence discussed in Section 4.2, may also include possible polarised endocytosis of matrix-degrading enzymes. For the cilium to function in this way, it must somehow interpret the direction mechanical forces are acting in order to direct these cellular responses in the appropriate spatial location. Farnum and Wilsman investigated whether this occurred through variation in the orientation of the ciliary axoneme with respect to the cell surface across the different zones of cartilage, and also in the position of the cilium on the cell surface [129]. Using EM analysis of equine knee articular cartilage, they found that, in the superficial zone, almost all cilia emerge from the side of the chondrocyte facing the underlying subchondral bone. In contrast, radiate zone cilia are directed towards the articular surface or subchondral bone in almost equal numbers, with zonal differences in orientation also observed. Interestingly, axonemal position and orientation were much more variable in non-loaded regions of cartilage, which exhibit less tissue organisation than loaded regions, indicating that mechanical forces determine both ciliary positioning and orientation, and matrix anisotropy. In human and murine growth plates, axonemal positioning similar to the radiate zone of articular cartilage has been observed within the proliferative and hypertrophic zones, which contain highly organised columns of cells that are generated by multiple rounds of cell division and subsequent movement (specifically rotation) of cells [130,131]. This positioning of cilia along the long axis of the tissue was disrupted in the growth plates of SMAD1/5 double knockout mice, which also exhibited loss of chondrocyte columns similar to that of KIF3A knockouts [17], possibly indicating a role for the cilium in cellular organisation of the growth plate. In light of these results, Farnum and Wilsman propose a model in which positioning of chondrocyte cilia may be established by the plane of cell division, which in turn is partly determined by processes involving ciliary proteins such as orientation of the mitotic spindle [132] and the planar cell polarity pathway (see below) [133].

However, it is currently unknown whether there is more than just correlation of cilia position and the generation of matrix anisotropy in the different zones of cartilage. The study of cilia in other tissues, particularly ones in which there is better understanding of the mechanism underlying polarised matrix secretion, could help to provide an answer to this question. One such tissue is tendon, in which collagen fibres are arranged in parallel bundles along the long axis of the tendon and thus along the line of mechanical stress. There is evidence that the highly organised tendon ECM is established during embryonic development through the action of plasma membrane protrusions known as fibripositors, which are also organised in parallel to the tendon axis. In embryonic chick tendon fibroblasts, Canty et al. observed that elongated membrane-bound Golgi-to-plasma membrane carriers (GPCs) transport newly-synthesised collagen molecules assembled into fibrils to fibripositors, which in turn facilitate deposition of fibrils into the developing tendon ECM [134]. Exactly how the cell orients these fibripositors parallel to the long axis of the tendon, and in turn determines the specific spatial pattern of collagen fibrils in the tendon ECM, is not known. Knockdown of the cell adhesion protein cadherin-11 in chick tendon cells resulted in loss of the collagen fibril-containing plasma membrane channels that run between cells along the tendon axis. This in turn was associated with a loss of parallel alignment of collagen in the matrix; however, this did not affect the presence of fibripositors, and thus cell-cell junctions may be involved in tissue organisation upstream of fibripositor positioning [135]. In contrast, inhibition of actin filament polymerisation resulted in loss of fibripositor-like structures and disrupted the parallel orientation of collagen fibril bundles in extracellular channels, indicative of a potential role for the actin cytoskeleton in tendon fibripositor formation and maintenance [136]. Further to this, Kapacee et al. found that tension is required for fibripositor formation and subsequent production of collagen fibrils with parallel orientation [137,138].

A possible hypothesis for how fibripositors are positioned on the tendon cell surface to facilitate this ordered deposition of collagen could be that this is linked to the positional information encoded by cilia, as proposed in cartilage. Primary cilia are aligned with parallel collagen fibrils in tendon [139], but this orientation is not affected by stress deprivation in rat tail tendon [140]. Also, isolated human tenocytes have random orientation of cilia, possibly due to lack of matrix in 2D culture, and thus this orientation could be due simply to constraints imposed by the collagen matrix [141]. In contrast, not much is known about cilia positioning in tendon. To begin investigating whether ciliary positioning affects fibripositor positioning, how cilia polarity is set up in the first place would need to be determined. There is growing evidence that this is related to the planar cell polarity (PCP) pathway, which enables the asymmetrical cellular features that generate cell polarity, such as the distribution of proteins in one specific cellular domain, to be aligned along one axis across an entire population of cells [142]. For example, mutation of PCP effectors Inturned and Fuzzy resulted in ciliogenesis defects that are linked to failure of organisation of the actin cytoskeleton that in turn controls the positioning of the basal body at the cell membrane [143], and other studies have made the link between PCP and cilia [133,144]. Supporting this, Goodyear et al. found that planar cell polarity genes such as *Ptk7* and *Vangl2* are required for production of the highly organised collagen matrix in the tectorial membrane of the cochlea, although reduction in ciliation through KIF3A knockdown resulted in no obvious changes in orientation [145]. Therefore, there is a need for experimental approaches that will enable alteration of cilia positioning directly (rather than just preventing ciliogenesis altogether) to investigate whether the cilium is involved in matrix organisation in tendon and other tissues.

## 6. Conclusions

In conclusion, many studies have found that there is an association between proteins required for assembling and maintaining a functional primary cilium, and the expression, secretion, and proteolytic remodelling of the extracellular matrix. A few studies have directly addressed how the ciliary machinery could regulate these processes, and the main mechanisms that have been proposed in the literature are summarised in Figure 1. However, there are still many aspects of these hypothesised mechanisms of regulation for which there is no experimental evidence. This is also true for the role of the cilium in organisation of the matrix across a tissue. These findings therefore highlight a need for investigation into how the ciliary machinery directly interacts with matrix-regulating processes in greater detail. Such research could provide useful insight into how disruption of ciliary proteins in disease causes matrix changes, which in turn could play an important role in disease pathogenesis.

## Figures and Tables

**Figure 1 cells-09-00278-f001:**
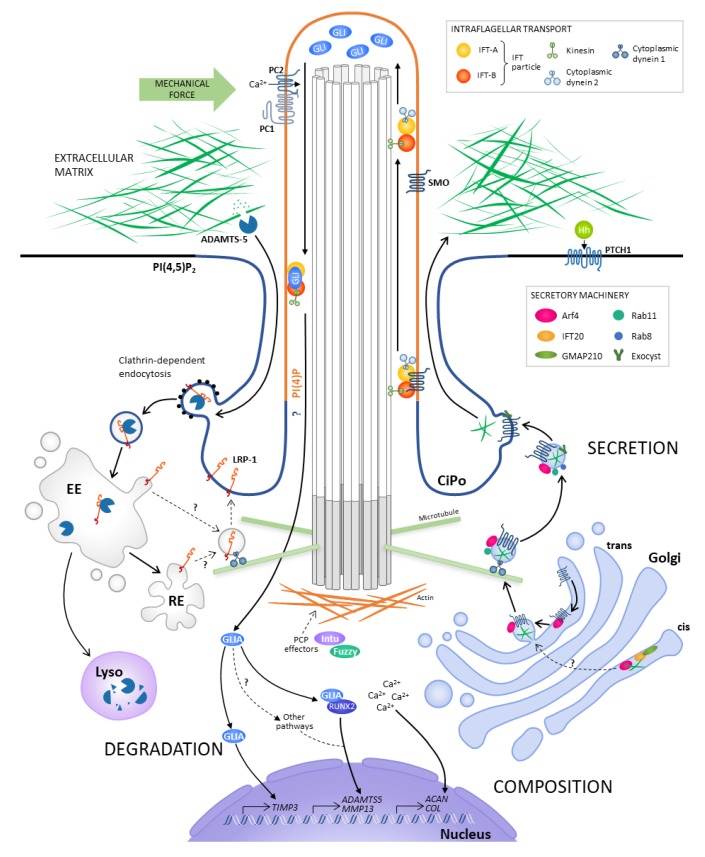
Hypothesised regulation of extracellular matrix composition by the cilium and ciliary proteins. Matrix-degrading proteases such as ADAMTS-5 bind to the LRP-1 receptor, which is polarised at the periciliary region in chondrocytes, possibly at the endocytosis-active structure, the ciliary pocket (CiPo). These enzymes are trafficked through the endosomal pathway to the lysosome (Lyso), where they are degraded. Recycling of the LRP-1 receptor to the ciliary/CiPo membrane may occur directly from the early endosome (EE) or via the recycling endosome (RE). Ciliary proteins have also been shown to regulate proteases at transcriptional level. The transcriptional effectors of the Hh pathway (GLIs), directly target genes encoding protease-regulating proteins such as TIMP-3, or interact with other transcription factors such as RUNX2 to regulate the expression of the protease itself. The Hh pathway could also interact with other pathways to regulate this process. Transcription of matrix components that determine the composition of the ECM occurs downstream of a calcium response triggered by the action of mechanosensitive receptors on the ciliary membrane, such as the polycystins PC1 and PC2. Proteins involved in intraflagellar transport (IFT), which mediates the trafficking of cargo from the cell body to the tip of the cilium and back via motor protein-driven IFT particles, have been linked to matrix secretion. IFT20 localises to the *cis*-Golgi, where it is anchored by the protein GMAP210, and could be involved in the targeting of proteins to the cilium via Arf4-driven trafficking and subsequent exocyst-mediated exocytosis. Exocytic and endocytic vesicles in the cytosol are transported by motor proteins along microtubules, which emanate from the main microtubule organising centre (MTOC) of the cell, the centrosome, at the base of the cilium. The mechanism by which IFT20 mediates this ciliary targeting, how it interacts with Arf4, and the extent of polarisation of matrix secretion at the cilium, are unknown. Positioning of the basal body at the plasma membrane, and therefore of the cilium and cellular processes that occur in the periciliary region, is influenced by the planar cell polarity (PCP pathway). PCP effectors such as Inturned and Fuzzy regulate the actin cytoskeleton, which in turn controls basal body polarisation.

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
