# Peer review of "Regulation of the Extracellular Matrix by Ciliary Machinery"

_cells, 2020, doi:10.3390/cells9020278_

Round 1
Reviewer 1 Report
Most studies of primary cilia focus on structural organization and roles in cellular signaling (ie how extracellular signals give rise to intracellular changes), and how both of these aspects are disrupted in ciliopathies. In this excellent review, Collins and Wann review the growing literature on how cilia affect the way that cells interact with and influence their environment. The generation and patterning of extracellular matrix is crucial for tissue organization and organogenesis. While the reviewed is mostly focused on bone and cartilage, this makes sense because of the importance of cell/ECM interactions in these tissues and since a subgroup of ciliopathies exhibit cartilage dysplasias, the link of cilia to ECM is tangible. The connection of Shh signaling to arthritis is exciting, and the final section on cilia, PCP, and ECM patterning was insightful. With minor textual revisions, this review should be published and I thank the authors for putting it together.
Line 85: Cartilage-specific knockout of Kif3A, which is a motor protein that transports IFT particles containing cargo from the cell body, is associated with altered type X collagen expression in the growth plate [16]. Reduced proteoglycan production and type X collagen expression are also observed upon deletion or mutation of proteins that form the IFT particles, such as IFT80 [17] and IFT88 [18].
Do any of the referenced articles (or the review authors) offer any speculation as to why absence or dysfunction of primary cilia due to IFT80/IFT88 mutations could lead to these changes?
Line 110: Mangos et al. showed that knockdown of PC1 during zebrafish development resulted in overexpression of collagens in different tissues, linking PC1 loss of function to expression of ECM components [26]. However, PC1 knockdown only slightly increased cyst formation, and collagen expression was increased in PC2-depleted embryonic kidneys, but diminished upon PC1 knockdown.
I find these two sentences hard to follow. In first sentence, reduced PC1 leads to increased collagen expression, but in second sentence, collagen expression… diminished upon PC1 knockdown. Maybe authors can rewrite or clarify this point.
Line 125-126: Instead, the group proposed that these results were due to altered expression of PC1 in ORPK chondrocytes.
“These results” refers to what exactly? If PC1 levels are experimentally reduced in a “compression-induced” sample, does aggrecan expression still increase? Has this been reported?
In Section 2, there are multiple mentions of “increase/decreased expression” of ECM components in response to changes in cilia structure/function. Has regulation of any of the genes encoding these components been directly linked to Hh signaling via GLI-mediated activation/repression? If this link is present in any of the cited references, it is worth highlighting.
Line 135: 3. Regualtion of Matrix Secretion
“Regulation” is misspelled.
Line 150: …fibres [38], possibly mediated by integrins later found to present on the ciliary membrane [39].
“to present” might be changed to “to reside” or “to be present” for clarity.
Line 156: 3.2. IFT20 has non-ciliary roles in secretion
In the studies reviewed in this section, emphasis is placed on intracellular trafficking role for IFT20, but there is no mention of the IFT20 ciliary phenotype. Are cilia present? If present, are they shorter?
IFT/cilia function and intracellular traffic function appear to be co-dependent, with roles for IFT20 in both, so is it feasible to isolate and assign roles in one function or the other?
Line 212. 3.3. Polarisation of secretion occurs at the immune synapse
In this section, parallels are drawn between membrane composition at the immunological synapse and the periciliary region of non-immune cells. How could IFT20 influence membrane composition? Most of evidence derives from studies on PIP kinases, phosphatases, and interactors – is there any study that shows mislocalization of these components when IFT20 is disrupted?
Line 251. 4. Regualtion of Matrix Degradation
“Regulation” is misspelled.
Line 318-319: However, direct stimulation of chondrocytes, in vitro or in explants, with Ihh does not result in increased matrix degradation or protease expression [77].
This statement contrasts the previous sentences, which describe studies using chondrocytes under cyclic tensile strain. Such strain can lead to deformation of the actin cytoskeleton, changes in nucleocytoplasmic trafficking, and even mechanical perturbation of the primary cilia, which all are considerations when interpreting this contrast. Might be mentioned in text.
Line 324: As the experiments in Thompson et al. were conducted…
Would be useful to insert reference here - Thompson et al. [77]
Line 342: Apart from this study, there is still very little direct, mechanistic evidence
Would be useful to insert a reference for clarity - Apart from this study [Ref?]
Line 372: ..which have increased protease activity in vitro…
What is assay here? Are detectable levels the same/different, but activity higher in the IFT88orpk chondrocytes? Please clarify.
Line 377: ….localises to Rab-11-postive…
“positive” is misspelled.
Line 457: …as described by Farnum and Wilsman.
Would be useful to insert a reference for clarity - as described by Farnum and Wilsman [114].
Line 556: …literature are summarised in figure 1…
Figure 1 should be capitalized.
Also, regarding Figure 1, because the final section of the review was dedicated to mechanisms behind spatial positioning of the ECM, it would be nice to see that referred to in the Figure. With some simple arrows and text referring to actin and planar cell polarity, with some minimal explanatory text in Figure legend, would be nice to see.
Author Response
Thank you for the detailed read and comments of our review, please find point by point responses below:
Most studies of primary cilia focus on structural organization and roles in cellular signaling (ie how extracellular signals give rise to intracellular changes), and how both of these aspects are disrupted in ciliopathies. In this excellent review, Collins and Wann review the growing literature on how cilia affect the way that cells interact with and influence their environment. The generation and patterning of extracellular matrix is crucial for tissue organization and organogenesis. While the reviewed is mostly focused on bone and cartilage, this makes sense because of the importance of cell/ECM interactions in these tissues and since a subgroup of ciliopathies exhibit cartilage dysplasias, the link of cilia to ECM is tangible. The connection of Shh signaling to arthritis is exciting, and the final section on cilia, PCP, and ECM patterning was insightful. With minor textual revisions, this review should be published and I thank the authors for putting it together.
Line 85: Cartilage-specific knockout of Kif3A, which is a motor protein that transports IFT particles containing cargo from the cell body, is associated with altered type X collagen expression in the growth plate [16]. Reduced proteoglycan production and type X collagen expression are also observed upon deletion or mutation of proteins that form the IFT particles, such as IFT80 [17] and IFT88 [18].
Do any of the referenced articles (or the review authors) offer any speculation as to why absence or dysfunction of primary cilia due to IFT80/IFT88 mutations could lead to these changes?
At the time these studies were conducted, and subsequently, the mechanism predominantly proposed to lead to these changes in KIF3A and IFT mutants was impaired Indian Hh signalling. Thus we included these studies this in the section concerning regulation of matrix phenotype by Hh signalling.
However, as Song et al. 2007 and McGlashan et al. 2007 both comment, there are changes to cell organisation and morphology, most notably orientation in the KIF3A mutant. This loss of orientation was also associated with activated focal adhesion kinase (FAK) and, in both KIF3A and ORPK mutant mice changes to actin cytoskeleton were observed, a link between ciliary proteins and the cytoskeleton that appears conserved across many cell types. It is still unclear how the role of ciliary-regulated Hh signalling might evolve in post-natal cartilage, particularly the adolescent growth plate and indeed in adult tissues more generally beyond the skeleton. Changes to skeletal development seen in both models are attributable to alterations in cell behaviour that go beyond altered Hh signalling. Indeed, when KIF3A was deleted, no change in Ptch1 mRNA was observed (Song et al., 2007), and when IFT88 was deleted (Haycraft et al., 2007) the authors point out not all elements of the resulting phenotype are seen in Hh mutants such as Ihh Gli3 mutants. Many studies have proposed tissue and cell patterning and associated matrix changes are also likely attributable to changes in wnt signalling including, non-canonical planar cell polarity (PCP). Exemplifying this, Yuan et al., 2015 attribute changes to faulty differentiation, due to reduced Hh signalling and increased wnt signalling. An earlier paper by Serra et al., 2013, proposes wnt is regulated downstream to Hh when IFT88 is deleted. We currently are investigating how mechanotransduction, mediated by ciliary IFT88 integrates with these factors and review ciliary mechanotransduction in the following section.
To bring a Hh-independent balance to this section we have now included comments to this end at the end of the section concerning changes to matrix phenotype and changed the title of this section to ciliary signalling regulates matrix phenotype.
Text changes and additions can be found line 96-124 in the revised manuscript.
Line 110: Mangos et al. showed that knockdown of PC1 during zebrafish development resulted in overexpression of collagens in different tissues, linking PC1 loss of function to expression of ECM components [26]. However, PC1 knockdown in kidney, where this causes cysts resulted in a decrease
I find these two sentences hard to follow. In first sentence, reduced PC1 leads to increased collagen expression, but in second sentence, collagen expression… diminished upon PC1 knockdown. Maybe authors can rewrite or clarify this point.
We agree that the phrasing was confusing, mainly because we did not clarify carefully enough that this paper shows that in some tissues, PC1 depletion increases the expression of collagens (e.g. increased Col2a1, Col9a2 and Col27a1 in the notochord) but in the embryonic kidney specifically, the expression of collagen (Col27a1) is reduced. The sentence has been rewritten to clarify this. Also, as this paper does not look at the phenotype of PC2-depleted kidneys, and since this section is focused on reviewing the evidence for a link between cystic kidney disease and changes in matrix, the part of the sentence about collagen expression results in PC2-depleted zebrafish has been removed.
Text changes can be found lines 141-149 in the revised manuscript.
Line 125-126: Instead, the group proposed that these results were due to altered expression of PC1 in ORPK chondrocytes.
“These results” refers to what exactly? If PC1 levels are experimentally reduced in a “compression-induced” sample, does aggrecan expression still increase? Has this been reported?
We have re-phrased the text and changes have been made to lines 162-171 in the revised manuscript.
In Section 2, there are multiple mentions of “increase/decreased expression” of ECM components in response to changes in cilia structure/function. Has regulation of any of the genes encoding these components been directly linked to Hh signaling via GLI-mediated activation/repression? If this link is present in any of the cited references, it is worth highlighting.
Thank you for raising this important point. To our knowledge none of the cited references include direct GLI-mediated activation or repression of matrix component genes. It is common place that Collagen X and aggrecan expression are altered but this is proposed to be via PTHrP or through synergism with other transcription factors. A study last year (Ali et al. 2019), which we now refer to in the manuscript, sought to identify GLI targets in neoplastic chondrocytes using GLI binding sites to identify Hh responsive genes. A supervised analysis of GLI1 targets related to signalling, transcription and extracellular matrix (ECM) contained a number of ECM-regulating factors. We have included reference to this study in the text (line 100) as it indicates the scope of potential matrix regulation either directly or indirectly downstream to ciliary Hh signalling, by testing for direct targets. It also exemplifies the complex, synergistic and interactive potential for regulation of matrix components directly, indirectly and partially by hedgehog which, potentially, could be a review in its own right if it were to go beyond the realms of this skeletal context. Nevertheless, where implicated as targets of Hh, we have sought to identify some of these for example Kopinke et al and/or added references (for example Horn et al 2012).
Line 135: 3. Regualtion of Matrix Secretion
“Regulation” is misspelled.
This has been corrected.
Line 150: …fibres [38], possibly mediated by integrins later found to present on the ciliary membrane [39].
“to present” might be changed to “to reside” or “to be present” for clarity.
We have corrected this error.
Line 156: 3.2. IFT20 has non-ciliary roles in secretion
In the studies reviewed in this section, emphasis is placed on intracellular trafficking role for IFT20, but there is no mention of the IFT20 ciliary phenotype. Are cilia present? If present, are they shorter?
This section is devoted to non-ciliary functions but most of these are proposed in the context of events upstream to ciliogenesis or in the cytoplasmic trafficking of proteins to the base of the cilium for axonemal trafficking/incorporation so the question is entirely valid and particularly in the sections related to IFT20 perturbations, we agree we haven’t mentioned the ciliary phenotype. Disruption of IFT20 in the kidney, retinal epithelia and in neural crest cells, as now described in the text, inhibits ciliogenesis and it is indispensable for ciliogenesis in condylar cartilage. This has been added to the relevant areas of the penultimate paragraph in this, now renamed (see below) section.
Text changes can be found lines 216-218, 250, and 258-261 in the revised manuscript.
IFT/cilia function and intracellular traffic function appear to be co-dependent, with roles for IFT20 in both, so is it feasible to isolate and assign roles in one function or the other?
We agree and on reflection we have re titled this section “IFT20; linking the golgi to cilia”. The only context where it is feasible to isolate them is perhaps the next section concerning the I.S.
Line 212. 3.3. Polarisation of secretion occurs at the immune synapse
In this section, parallels are drawn between membrane composition at the immunological synapse and the periciliary region of non-immune cells. How could IFT20 influence membrane composition? Most of evidence derives from studies on PIP kinases, phosphatases, and interactors – is there any study that shows mislocalization of these components when IFT20 is disrupted?
Yes we felt useful, as others have before us, to draw comparisons between the cilium and the I.S using IFT20 as a focus to bridge from section prior. We do not know of any direct evidence showing a role for specifically IFT20 in directly modulating membrane composition or the modulators as suggested, to avoid confusion we have changed the bridging text at the start of this section. It is likely, to our mind, that any perturbation to ciliary assembly or indeed trafficking and recycling in the peri-ciliary region or I.S, would likely disrupt the localisation of PIP kinases and other effectors. Kosling et al. 2018 proposed that ciliary localisation of INPP5E inside the cilium is IFT dependent. This would likely effect ciliary axoneme membrane composition. It is far from well understood if the pocket region around the cilium has a unique composition or would be indirectly effected by this failure of ‘innerciliary’ trafficking. We have now added text to this effect towards the end of this section.
Text changes can be found in the revised manuscript lines 309-319.
Line 251. 4. Regualtion of Matrix Degradation
“Regulation” is misspelled.
This has been corrected.
Line 318-319: However, direct stimulation of chondrocytes, in vitro or in explants, with Ihh does not result in increased matrix degradation or protease expression [77].
This statement contrasts the previous sentences, which describe studies using chondrocytes under cyclic tensile strain. Such strain can lead to deformation of the actin cytoskeleton, changes in nucleocytoplasmic trafficking, and even mechanical perturbation of the primary cilia, which all are considerations when interpreting this contrast. Might be mentioned in text.
We agree entirely with this comment and have added to the text accordingly.
Text changes can be found 396-403 in the revised manuscript.
Line 324: As the experiments in Thompson et al. were conducted…
Would be useful to insert reference here - Thompson et al. [77]
We agree and have done so.
Line 342: Apart from this study, there is still very little direct, mechanistic evidence
Would be useful to insert a reference for clarity - Apart from this study [Ref?]
We agree and have done so.
Line 372: ..which have increased protease activity in vitro…
What is assay here? Are detectable levels the same/different, but activity higher in the IFT88orpk chondrocytes? Please clarify.
We have added text to include details on assay, which used co-culture of purified aggrecan in vitro coupled with probing of aggrecan neoepitopes to assess activity. We have also added text to clarify expression levels of protease expression in this section.
Text changes can be found lines 452 to 456 in the revised manuscript.
Line 377: ….localises to Rab-11-postive…
“positive” is misspelled.
This has been corrected.
Line 457: …as described by Farnum and Wilsman.
Would be useful to insert a reference for clarity - as described by Farnum and Wilsman [114].
We have added as suggested.
Line 556: …literature are summarised in figure 1…
Figure 1 should be capitalized.
This has been corrected.
Also, regarding Figure 1, because the final section of the review was dedicated to mechanisms behind spatial positioning of the ECM, it would be nice to see that referred to in the Figure. With some simple arrows and text referring to actin and planar cell polarity, with some minimal explanatory text in Figure legend, would be nice to see.
We agree that the concepts discussed in all other sections of the review have been reflected in the schematic except the section on matrix organisation. Representations of the microtubule network, actin, and PCP effectors such as Inturned and Fuzzy, have now been included in the figure. The figure legend has also been amended to include a sentence explaining the link between the PCP pathway, actin remodelling and basal body positioning.
Reviewer 2 Report
The manuscript by Collins and Wang provides an extensive review of the role of the ciliary machinery in the regulation of the extracellular matrix, most notably in cartilage formation and homeostasis.
Although potentially useful it is hampered by the focus on chondrocytes, as cilia are present on numerous other cells in (polarized) tissues, predominantly in epithelia and endothelia. Here, the cilia extend apically into cavities or lumina and act mainly as mechanosensor in fluid surroundings. The secretion of matrix constituents in these polarized tissues is mainly at the basal side. Although a ciliary pocket is also present including relevant parts of the inherent machinery, the function here must be principally different. Centrally is the question whether the cilium is always present and functional in the described situations. It is described that in stressed circumstances (high or disturbed flow versus low laminar flow) the cilium is usually shortened or even absent, thereby regulating gene expression patterns (Egorova et al Differentiation 2012, Van der Heiden et al Atherosclerosis 2008). The authors do not allude to this aspect in their review. Subramanian (Int J Biochem Cell Biol 2017) also describes mechanical loading in chondrocytes associated with the length of the cilium, not focusing on secretory processes.My concern is that the ciliary domain including pocket during mechanosensing (see also line 436) and matrix regulation are quite different and in the latter case might even lack the cilium.
Line 25. “the older of the centrioles” introduces seemingly a paradox, as the younger one will become the older after mitosis. In between the transition something happens to make only one centriole as basis for the basal body. Please explain.3.Line 45. “migration, proliferation and differentiation” are principal phenomena in embryonic development, yet no mention is made on developmental phases starting as early as the primitive node and notochordal plate with motile and non-motile cilia determining e.g. left/right asymmetry (Poelmann, Anat Embryol 1981, Sulik et al Dev Dyn 1994). Many more examples can be found and in my opinion this review gains strength by adding a section on these matters.
Early in the manuscript a reference should be made to Figure 1 and the major players should be described. Also the involvement of the cellular actin and microtubular systems could be included, see also line 425 and ref #98. Correct the titles in lines 135 and 251. Chapters 3.2 and 3.3 provoke me with the suspicion that the “specialized membrane patches” (line 238) are just that, incidentally the cilium is only part of one such specialized area (see also my last line under #1). Please comment on this. The section on the immune synapse (line 214 -217) seems an oversimplification as ref #53 is more careful in the conclusions than the current authors. Please read again ref#53.8.line 257. Does this concern the function of the cilium as stated or the centriole/basal body and adjoining domain?
9 line 322, ad ref # after Lin et al.
Line 457, add ref # after Farnum and Wilsman.11 line 468, the actin system has been alluded to, but the microtubular connection has not been mentioned.
Author Response
Thank you for the thoughtful comments related to our manuscript which we have considered and used to improve the review as outlined in a point to point response shown below:
The manuscript by Collins and Wang provides an extensive review of the role of the ciliary machinery in the regulation of the extracellular matrix, most notably in cartilage formation and homeostasis. Although potentially useful it is hampered by the focus on chondrocytes, as cilia are present on numerous other cells in (polarized) tissues, predominantly in epithelia and endothelia.
Here, the cilia extend apically into cavities or lumina and act mainly as mechanosensor in fluid surroundings. The secretion of matrix constituents in these polarized tissues is mainly at the basal side. Although a ciliary pocket is also present including relevant parts of the inherent machinery, the function here must be principally different. Centrally is the question whether the cilium is always present and functional in the described situations.
We entirely agree and are aware this review has a bias towards skeletal cells which we sought to justify. This was, as stated, largely due to weight of evidence for ciliary regulation of matrix being pre-dominantly related to musculoskeletal biology but as picked up on by other reviewer is also potentially most relevant in this context, due to the skeletal ciliopathies. That said, we agree we could have worked harder to give broader context at times and have sought to do this now throughout, adding references from beyond skeletal cells in the contexts highlighted by both reviewers.
We agree entirely that in circumstances where the cilium is not assembled or positioned to directly interact with the matrix, many of the concepts we have covered or proposed will not be relevant. We have tried to consider some of these contexts (e.g the immune synapse) where we felt it helped consider working hypotheses proposed within the review. We have now added to the text the critical points the reviewer makes here with respect to polarised epithelial ciliogenesis and basal-lateral matrix secretion at the beginning of matrix organisation section.
Text changes can be found lines 520-526 in the revised manuscript.
It is described that in stressed circumstances (high or disturbed flow versus low laminar flow) the cilium is usually shortened or even absent, thereby regulating gene expression patterns (Egorova et al Differentiation 2012, Van der Heiden et al Atherosclerosis 2008). The authors do not allude to this aspect in their review.
Subramanian (Int J Biochem Cell Biol 2017) also describes mechanical loading in chondrocytes associated with the length of the cilium, not focusing on secretory processes.
My concern is that the ciliary domain including pocket during mechanosensing (see also line 436) and matrix regulation are quite different and in the latter case might even lack the cilium
Indeed whilst we have sought to address non-ciliary mechanisms in this review we have not alluded to situations in potentially ciliated cells such as vascular endothelium that are not ciliated as a result of environment. We have now added this important point, related to situations where the cilium is compromised or altered by its mechanical environment to the end of the section related to mechanical regulation.
Text changes can be found lines 184-191 in the revised manuscript.
Line 25. “the older of the centrioles” introduces seemingly a paradox, as the younger one will become the older after mitosis. In between the transition something happens to make only one centriole as basis for the basal body. Please explain.
We have improved this brief introduction to include the main processes in centriole to basal body transition supporting ciliogenesis.
Text changes can be found lines 25-30 in the revised manuscript.
3.Line 45. “migration, proliferation and differentiation” are principal phenomena in embryonic development, yet no mention is made on developmental phases starting as early as the primitive node and notochordal plate with motile and non-motile cilia determining e.g. left/right asymmetry (Poelmann, Anat Embryol 1981, Sulik et al Dev Dyn 1994). Many more examples can be found and in my opinion this review gains strength by adding a section on these matters.
The focus of this review was to consider ciliary regulation of extracellular matrix, not least as the opposite direction was nicely written about in 2012 by Segger-Nukpezah & Golemis. We are of course aware of the fundamental roles the primary cilium plays in early development. Clearly, in the context of migration, the matrix substrate is a key driver but the proposed roles of cilia in these processes are predominantly thought to be in the context of reception of migratory cues. Similarly, the most prominently researched roles for the primary cilium in the node and left/right asymmetry focus on perception of nodal flow generated by motile cilia. This is clearly a fundamental role for the ciliary machinery that does lie upstream to differentiation, patterning and matrix regulation but could not find primary research describing, even indirectly, the inevitable matrix changes downstream to this. There are descriptions of mechanisms underpinning ECM remodelling, such as the transcription factor HAND2 regulating matrix proteases in zebrafish gut-looping morphogenesis but we could not link this downstream to the primary cilium.
Early in the manuscript a reference should be made to Figure 1 and the major players should be described.
We have made changes throughout, to make better reference to Figure 1 when they are first described in the text.
Also the involvement of the cellular actin and microtubular systems could be included, see also line 425 and ref #98.
We have now included into the figure as suggested.
Correct the titles in lines 135 and 251.
These are now correct.
Chapters 3.2 and 3.3 provoke me with the suspicion that the “specialized membrane patches” (line 238) are just that, incidentally the cilium is only part of one such specialized area (see also my last line under #1). Please comment on this.
It is certainly true that in a vast array of contexts, beyond just the specialisation of the ciliary membrane or hypothesised specialisation of the pocket, cells create areas of membrane with different lipid make-ups. We did not aim to convey that matrix regulation, secretion, or organisation are only regulated at the cilium but to focus on ciliary-organised mechanisms or scenerios with similarities such as the I.S, where researchers themselves have used this quoted term. We have been clear to cite this in the manuscript now.
The section on the immune synapse (line 214 -217) seems an oversimplification as ref #53 is more careful in the conclusions than the current authors. Please read again ref#53.
Upon re-reading of ref 53 (Stinchcombe et al., 2015), we agree that this is an oversimplification of the findings. The only stage of immune synapse formation which the authors state is similar to that of ciliogenesis, is polarisation of centrosome at the plasma membrane of the immune synapse, rather than the entire process of immune synapse formation being similar to ciliogenesis. The text has been amended to reflect this.
Text changes can be found lines 277-281 in the revised manuscript.
8.line 257. Does this concern the function of the cilium as stated or the centriole/basal body and adjoining domain?
The section that follows this sentence includes discussion of how both the cilium and the periciliary environment containing the centriole/basal body and adjoining membrane, regulate matrix-degrading proteases. Specifically, the cilium and ciliary signalling (Hh) are discussed in the context of regulation of protease transcription, and the periciliary membrane domain is discussed in the context of regulation of protease activity via endocytosis. [The word “synthesis” has also been changed to “expression” in this sentence, as the former also alludes to secretion, and there is no evidence to our knowledge that Hh signalling is involved in secretion of proteases.
9 line 322, ad ref # after Lin et al.
This has been corrected
Line 457, add ref # after Farnum and Wilsman.
This has been corrected
11 line 468, the actin system has been alluded to, but the microtubular connection has not been mentioned.
This has now been amended as the microtubular network, carrying vesicles, is depicted in Figure 1 and referred to in the revised legend.
Round 2
Reviewer 2 Report
The authors have adapted the text as required.